


# AutoATES v2.0: Automated avalanche terrain exposure scale mapping

Håvard B. Toft[1,2]*, John Sykes[3,4], Andrew Schauer[4], Jordy Hendrikx[5,6,2] and Audun Hetland[2]

[1] Norwegian Water Resources and Energy Directorate, Oslo, Norway

[2] Center for Avalanche Research and Education, UiT the Arctic University of Norway, Tromsø, Norway

[3] SFU Avalanche Research Program, Department of Geography, Simon Fraser University, Canada

[4] Chugach National Forest Avalanche Center, Girdwood, AK, USA

[5] Antarctica New Zealand, Christchurch, New Zealand

[6] Department of Geosciences, UiT the Arctic University of Norway, Tromsø, Norway

*Corresponding author address:*

Håvard B. Toft, Norwegian Water Resources and Energy Directorate, Oslo, Norway; tel:
+47 454 82 195; email: htla@nve.no

Keywords: ATES, GIS, snow, avalanche,

*Throughout the paper, we will use the terms: model and algorithm interchangeably, but they convey the same meaning.*

## Abstract

This paper documents substantial improvements to the original automated avalanche terrain exposure mapping (AutoATES v1.0) algorithm. The most significant drawbacks of AutoATES v1.0 have been addressed by including forest density data, improving the avalanche runout estimations in low-angle runout zones, accounting for overhead exposure and open-source software. The algorithm also supports the new ATES v2.0 terrain class 'extreme' terrain. We used two benchmark maps from Bow Summit and Connaught Creek to validate the improvements from AutoATES v1.0 to v2.0. For Bow Summit, the F1 score (a measure of how well the algorithm performs) improved from 64.01% to 77.30%. For Connaught Creek, the F1 score improved from 39.81% to 71.38%. The main challenge limiting large-scale mapping is the determination of optimal input parameters for different regions and climates. In areas where AutoATES v2.0 is applied, it can be a valuable tool for avalanche risk assessment and decision. Ultimately, our goal is for AutoATES v2.0 to enable efficient, large-scale, and potentially global ATES mapping in a standardized manner rather than based solely on expert judgement.

## 1. Introduction

Approximately 140 fatal accidents result from snow avalanches in Europe and Northern America annually (Techel et al., 2016, 2018; Birkeland et al., 2017). In recent decades, most of these fatalities have been related to the recreational use of avalanche terrain (Engeset et al., 2018). In 90% of cases, the victim or someone in their group triggered the avalanche (Schweizer and Lütschg, 2001). Many countries have established an avalanche forecasting service to increase awareness of and help mitigate the risk of avalanches and focus on increased public education (Engeset et al., 2018). Despite the availability of public regional avalanche forecasting in many countries, assessing the avalanche risk is a complex task for backcountry recreationists due to the spatial and temporal variability of snow. This results in a wicked learning environment, where feedback is not always reliable (Fisher et al., 2022). The avalanche risk is managed by performing detailed assessments of, i.e., weather, snowpack, and signs of instabilities at a regional scale. An efficient method to mitigate the avalanche hazard is using appropriate terrain for the avalanche conditions (Thumlert and Haegeli, 2017).

Assessing avalanche terrain may be intuitive for avalanche professionals (Landrø et al., 2020), however, this may not be the case for recreational users of avalanche terrain. The avalanche terrain exposure scale (ATES) is a terrain classification system developed by Parks Canada to communicate the complexities and risks of traveling in avalanche-prone terrain (Statham et al., 2006). ATES is a commonly used classification scheme



worldwide and quantifies the avalanche terrain into an easy-to-understand rating: simple (class 1), challenging (class 2), and complex (class 3) terrain. A more detailed technical description of these classes is presented in Statham et al. (2006) and also reproduced in Larsen et al., (2020). Recently, the ATES classification scheme has been updated to include two additional ratings; non-avalanche (class 0, optional) and extreme (class 4) terrain to complement the current ATES classes from 1-3 (Statham and Campbell, 2023).

Avalanche hazard mapping has been common practice for decades to calculate the potential consequence of different avalanche scenarios related to infrastructure (Schläppy et al., 2013). The maps are often calculated for a specific return period (i.e., the probability of a given magnitude avalanche every 100 years) and determines the likelihood of an avalanche (sometimes with specific impact pressures) within a defined area. The return periods vary by application, and by country (DIBK, 2017; BFF and SLF, 1984). In recent years, it has become more common to undertake an assessment of avalanche terrain zoning, where the aim is to divide the avalanche terrain into different zones or classes (e.g., ATES) depending on a specific skill level (CAA, 2016) or for recreational purposes (Campbell and Gould, 2013; Schmudlach and Köhler, 2016; Thumlert and Haegeli, 2017; Harvey et al., 2018; Larsen et al., 2020; Schumacher et al., 2022). In addition to mapping to inform users, ATES mapping has also been used as an important component to assess and measure terrain use preferences of backcountry users using GPS at a range of spatial scales (e.g., Hendrikx et al., 2022; Johnson & Hendrikx, 2021; Sykes et al., 2020). Statham et al. (2006) noted that the ultimate goal would be to apply the ATES classification spatially to produce ATES maps across entire regions. From 2009 through 2012, Avalanche Canada mapped several thousand square kilometers of avalanche terrain (Campbell and Gould, 2013). This mapping was undertaken using a combination of manual mapping and GIS-assisted mapping workflows, which relied heavily on expert judgement. As part of this work, Campbell and Gould (2013) identified the need for a more quantifiable model and suggested a new zonal ATES model for GIS-assisted classification. Therefore, the majority of large-scale mapping of ATES have been limited by the manual labor needed to generate maps. ATES is, therefore, typically only available in high-use areas due to the number of resources needed to generate ATES maps.

The first attempt at a fully automated ATES classification was made by Larsen et al. (2020) using a combination of the zonal and technical model of ATES (Campbell and Gould, 2013; Statham et al., 2006). The authors developed an automated ATES (AutoATES v1.0) algorithm that produces spatial ATES maps for all of Norway, using only a digital elevation model (DEM) as input. The main limitations of this work were that the algorithm did not account for forest data, or overhead exposure, and the performance of the simple avalanche runout simulation was insufficient in flat runouts. The algorithm was also heavily dependent on proprietary software (Larsen et al., 2020), thereby increasing the monetary and computing costs to operate the model, and also limiting open sources access.

In avalanche terrain zoning, the main goal is to divide the terrain into different zones or classes representing different areas of hazard, using a defined classification scheme. Avalanche terrain, especially when complex is the result of interactions of multiple release areas, tracks, and deposition areas. Within these three areas, other factors, i.e., terrain traps or forest density, could make terrain management more complex due to a more severe outcome. The two most important components in making a good avalanche terrain zoning algorithm are the delineation of the start zone area, as defined by the potential release area (PRA) model, and the avalanche runout distance and width, accounting for the track and deposition area. An increase in accuracy in either of these components directly benefits avalanche terrain zoning models. Additional factors like forest density have also been found to be significant (Delparte, 2008; Schumacher et al., 2022).

The use of an appropriate PRA model to delineate the start zones of avalanche paths, is critical when creating a good avalanche terrain zoning model (Sykes et al., 2022). The PRA establishes the baseline for where avalanches may release and is used as an input for the avalanche runout simulations. Manual classification of PRAs is time-consuming and often involves field observations, historic events review, and numerical



simulations (Bühler et al., 2018). A range of different PRA algorithms based on GIS or remote sensing have been developed (Bühler et al., 2018, 2013; Maggioni and Gruber, 2003; Barbolini et al., 2011; Pistocchi and
Notarnicola, 2013; Chueca Cía et al., 2014; Andres and Chueca Cia, 2012; Ghinoi and Chung, 2005; Veitinger et al., 2016).

The two most commonly used PRA algorithms are those developed by Bühler et al. (2013) and Veitinger et al. (2016). A key difference between the two algorithms is that the one from Bühler et al. (2013) produces a
binary polygon-based output, while the one from Veitinger et al. (2016) produces a continuous raster layer ranging from 0 to 1. Both algorithms are considered to have a good performance, even though the polygon-based algorithm was found to be slightly more accurate (Bühler et al. 2018). In prior automated ATES mapping work, Larsen et al., (2020) used the PRA algorithm of Veitinger et al. 2016 for the AutoATES v1.0 algorithm due to the continuous raster output. It is possible to include a binary forest parameter in the
Veitinger et al (2016) PRA model. However, the binary nature of the parameter results in coarse output, as the model removes all PRAs when the forest parameter takes the value 1. Sharp (2018) improved this PRA algorithm by incorporating forest density as a parameter in the fuzzy logic operator, making the forest interaction more dynamic.

There are several avalanche runout simulation models available which, given specific start zone inputs from the PRA model, outputs the potential track and deposition area. In principle, these runout models could be divided into two categories: (1) process-based, which attempt to calculate all the physical properties involved, or (2) empirical models which is driven by data-based observations. Which modelling approach to choose depends on the problem to be solved, data availability, the required accuracy and the spatial scale
(D'Amboise et al., 2022). Given access to highly detailed data and unlimited computational power, the process-based models outperform the data-based empirical models. However, given the limitations in computational power when processing large areas and the need for more accurate DEMs in many countries, the data-based model is more suitable for large-scale mapping applications.

Two of the most common process-based simulation tools for avalanche hazard assessment are the RAMMS (Christen et al., 2010) and Samos-AT (Sampl and Zwinger, 2004) models. Both models are made to simulate an accurate prediction of avalanche runout distances, flow velocities and impact pressures in a 3-dimentional space. These models are typically calibrated towards known avalanches with long return periods and defines potential avalanche terrain. These models are suitable for avalanche terrain zoning, where the aim is to divide
the potential avalanche terrain into different zones, across large spatial areas such as regional forecast areas or entire countries, these models are less suitable.

In contrast to the process-based models, data-based models are computationally inexpensive and can more easily be applied to large geographic areas. A common data-based method to delineate avalanche runout is
applying the classical runout angle concepts and path routing in three-dimensional terrain (D'Amboise et al. 2022).

In prior automated ATES mapping work, Larsen et al. (2020), used the multiple flow direction algorithm D-infinity (Tarboton, 1997). This algorithm is coupled with the travel angle (i.e., alpha angle). The D-infinity
algorithm identifies the cells downslope of the starting cell for each PRA cell. The algorithm spreads downslope until a defined alpha angle is reached from the starting cell (as per Heim, 1932; Lied & Bakkehøi, 1980; Toft et al., 2023). While used in hydrology applications, a substantial weakness of the D-infinity algorithm is that it cannot appropriately model avalanche movement, which may occasionally flow in flat and uphill terrain.
Recently, D'Amboise et al. (2022) presented a new customizable simulation package (Flow-Py) to estimate the runout distance and intensity of avalanches. The model utilizes persistence-based routing instead of terrain-based routing, enabling the simulation to respond appropriately to flat or uphill terrain. Where the





D-infinity algorithm only considers flow direction, the Flow-Py algorithm also considers flow process intensity. They use the same stopping criteria to estimate the runout distance by defining the alpha angle from the initial starting cell.

## 2. Model motivation

The main objective of the AutoATES v2.0 algorithm is to improve large-scale spatial ATES mapping, update the mapping to reflect recent changes in ATES which include the two new terrain classes (0 and 4), and improve the model workflow. Manual ATES classification using avalanche experts is time-consuming and expensive (Sykes et al., 2020), limiting large-scale mapping. For AutoATES v2.0 to be a viable option for large-scale ATES classification, the model performance should, on average, be as accurate as manual mapping or better.

### 2.1 Model description

This paper aims to document the improvements made to the AutoATES v1.0 algorithm initially developed by Larsen et al. (2020). In AutoATES v2.0, the influence of the forest density has been included by integrating the parameter into the PRA model (as per Sharp, 2018), track and the deposition area. The TauDEM runout model (Tarboton, 2005), which is known to perform poorly in flat deposition areas (Larsen et al. 2020), has been replaced by the new Flow-Py model (D'Amboise et al. 2022). Another advantage of the Flow-Py model is a separate output layer which enables the model to quantify the overhead exposure from multiple avalanche paths, which is an important consideration in the updated ATES model. Finally, the model now also includes the new ATES class for extreme terrain (Statham and Campbell, 2023) and steps to improve delineation of terrain traps.

### 2.2 Implementation

To secure a broad adaptation of the new AutoATES model it is important that the model is open-source and easy to use. The v1.0 algorithm was written using proprietary software. We have resolved this by rewriting the entire v2.0 algorithm into the programming language Python using widely available and open-source modules. The AutoATES v2.0 model is available on GitHub (Toft, Sykes, et al., 2023).

### 2.3 Input data

The minimum input data required to run the full AutoATES v2.0 is a DEM and forest density raster, both using the GeoTIFF format. It is also possible to run the algorithm with only a DEM as input, but the output would then only be valid for open, non-vegetated terrain. Both rasters must have a matching spatial resolution, extent and be defined using a projected coordinate system. The algorithm has been tested with spatial resolutions ranging from 5 to 30 m (cell sizes), but it should be possible to run other spatial resolutions.

Our parametrization for forest density allows for various metrics of forest density inputs. The algorithm is designed to work with stem density, percent canopy cover, basal area and no forest (only for mapping of open terrain). The forest type must be defined using a string in the beginning of the Python script ('stems', 'pcc', 'bav' and 'no_forest'). Forest density influences snow accumulation and snowpack stability, with denser forests generally reducing the risk of avalanches.

#### 2.3.1 Percent canopy cover

Canopy cover has a direct relationship with radiation balance and can impact formation of persistent weak layers as well as give an estimate of the degree of snowfall intercepted by trees prior to falling onto the snowpack. Percent canopy cover is a widely used metric that quantifies the extent of forest density by measuring the proportion of the ground area obscured by tree canopies when viewed from above. Percent canopy cover can be estimated using various methods, including aerial photography, satellite imagery, remote sensing techniques, and ground-based measurements. The resultant parameter used in our model has a value ranging from 0 to 100.

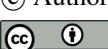




### 2.3.2 Stem density

Stem density is a metric used to quantify the number of tree stems (trunks) per unit area, typically expressed as stems per hectare or stems per square meter, which provides insight into forest structure and composition. Stem density can influence the snowpack stability and avalanche initiation, as a higher stem
density generally results in more trees obstructing and anchoring the snow, thereby reducing the likelihood of avalanche occurrence. Stem density can be measured through various techniques, including field surveys, aerial imagery analysis, or remote sensing data. The resultant parameter used in our model can have a value ranging from 0 to infinity, and is stated in stem density per hectare.

### 2.3.3 Basal area

The basal area represents the total cross-sectional area of all living trees in the dominant, co-dominant, and high intermediate crown positions and is measured in $m^2$/hectare (Sandvoss et al., 2005). The advantage over crown cover and stem density is that it incorporates the size of trees in addition to the number of trees and is a more direct measurement of the density of the forest vegetation. The resultant parameter
used in our model can have a value ranging from 0 to infinity, and is stated in $m^2$ per hectare.

### 2.4 Model components

The AutoATES v2.0 algorithm is split into two main components: (1) pre-processing and (2) the AutoATES classifier. In the pre-processing step, the DEM and forest density rasters are used as input for the start zone
PRA algorithm. When the PRA calculations are complete, the PRA and DEM is used to calculate the avalanche runout using the Flow-Py component. When all the key components are calculated, they are used as input for the AutoATES classifier which assigns the final ATES classes for each raster cell (Figure 1).

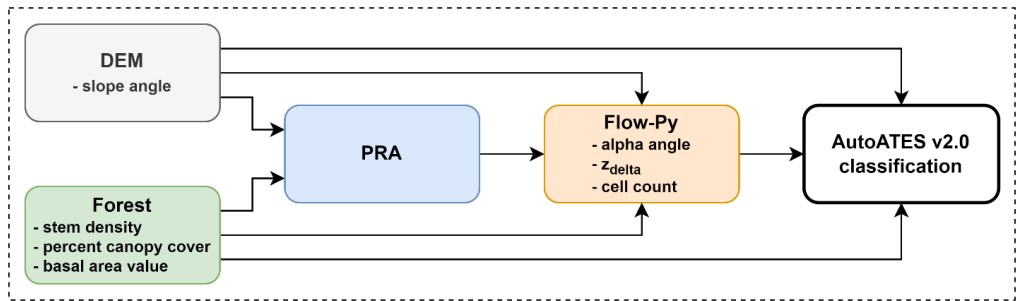

Figure 1: The main components of the AutoATES v2.0 algorithm. First, a pre-processing step is completed to calculate all the necessary raster layers using PRA and Flow-Py. Finally, the AutoATES classifier is used to assign the final ATES classifications.

### 2.4.1 PRA

The AutoATES v1.0 algorithm (Larsen et al., 2020) incorporated the PRA model developed by Veitinger et al.
(2016) to calculate the potential release areas. This PRA model (v1.0) uses slope angle, roughness and windshelter as input parameters. Sharp (2018) modified this algorithm to also include forest density. The algorithms utilize Cauchy membership values to assign the importance of each parameter (Jang et al. 1997). A Cauchy membership values must be defined for each input variable (Eq. 1).

$$\mu(x) = \frac{1}{1+\left(\frac{x-c}{a}\right)^{2b}} \qquad (1)$$

where $\mu(x)$ is the Cauchy membership value, x is an input variable, and a, b, and c are parameters which control the weight of each input variable. We use the membership values suggested by Veitinger et al. (2016)




for slope angle and windshelter, while using the value suggested by Sharp (2018) for stem density (Figure 2).
In our modified version of the PRA model (v2.0), we have chosen to remove the roughness parameter due to
the scale issues with 5-30 m cell sizes. The removal of roughness makes it less ideal for higher resolution
DEMs (< 5 m cell sizes). We also defined some new membership functions based on input from Parks Canada
avalanche experts and through testing of the AutoATES model on our two study areas. These values could be
fine-tuned for different inputs to improve the performance of the PRA model.



Figure 2: The different Cauchy functions used by Veitinger et al. (2016) and Sharp (2018) for slope angle and stem
density. We have suggested new membership values for windshelter, canopy cover (%) and basal area. We
recommend that these values are fine-tuned for specific datasets and applications, read a more in-depth discussion of
this in section 4.3.

The Cauchy membership values from slope angle, windshelter and forest density is used as inputs for the
fuzzy operator. We use the same "fuzzy AND" operator used by both Veitinger et al. (2016) and Sharp (2018),
originally defined by Werners (1988). The PRA value is therefore defined as follows in Eq. 2:

$$\mu_{PRA}(x) = \gamma \cdot \min(\mu_s(x), \mu_w(x)\mu_f(x)) + \frac{(1-\gamma)+(\mu_s(x),\mu_w(x)\mu_f(x))}{3}, \tag{2}$$

$$x \in X, \gamma \in [0,1]$$

With three fuzzy sets slope angle $\mu_s(x)$, windshelter $\mu_w(x)$, forest density $\mu_f(x)$ and with $\gamma$ defined in Eq. 3
as:

$$\gamma = 1 - \min(\mu_s(x), \mu_w(x)\mu_f(x)) \tag{3}$$

The PRA output is a continuous layer ranging between 0 (not likely) to 1 (very likely). Most data-based runout
models need release areas in a binary format where 0 is no potential release areas, while the potential release
areas are encoded as 1. To convert the PRA layer to a binary format, we select a cut off threshold (PRA$_{threshold}$)
where all pixels above this value is considered a potential release area for the runout modelling. We found
the PRA$_{threshold}$ from Larsen et al. (2020) to be too conservative and have therefore increased the value to
0.15. The PRA$_{threshold}$ could be adjusted depending on whether frequent or more extreme avalanche scenarios
are of interest.

We have also adjusted how the windshelter index is calculated. Using a 2m DEM, Veitinger et al. (2016)
resampled the DEM by a factor of 5 (from 2m to 10m) and applied a 11x11 sliding window. This is according
to the recommendations of Plattner et al. (2006) which found the optimal radius to be 60 meters, followed
by a secondary optimal radius of 250 meters. To achieve the same results, we removed the down sampling
factor of 5 and used the 10m DEM directly to calculate the windshelter index. If other DEM resolutions are
to be used, the windshelter index should be adjusted accordingly to use either 60 m (recommended) or 250



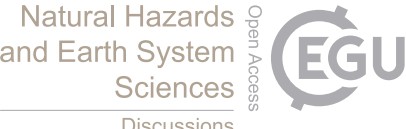

m as the radius around each cell. This could be done by either resampling the spatial resolution or changing the size of the sliding window.


**2.4.2 Avalanche simulation**

The Flow-Py model developed by D'Amboise et al. (2022) is used for the avalanche simulation of the potential track and deposition area. It is similar to the TauDEM algorithm utilized in AutoATES v1.0 which uses the alpha angle to limit the flow (Larsen et al., 2020; Tarboton, 1997). Flow-Py also includes a flow process intensity parameter which makes it able to handle mass movement in flat and uphill terrain, significantly improving the output compared to the previous model. Another advantage with the FlowPy model is the additional output layers which represents the overhead exposure. We utilize the cell count and $z_{delta}$ layer by scaling the two layers and taking their average value which represents the overhead exposure layer. In the AutoATES v2.0 algorithm it is possible to select cell count, $z_{delta}$ or both to represent the overhead exposure. The layer enables us to quantify the exposure from different release areas at every raster cell. We use the forest detrainment module of Flow-Py which makes it possible to use forest density as an input layer to limit spreading and runout distance. An in-depth description of the Flow-Py simulation package can found in D'Amboise et al. (2022).

**2.4.3 AutoATES classifier**

When the pre-processing of PRA and Flow-Py is completed, the AutoATES classifier uses a set of map algebra equations to define each ATES class. The following raster layers from the pre-processing step are used as input in the AutoATES classifier:

- Slope angle (calculated from the DEM)
- Forest density (provided by the user, as per section 2.3.1-2.3.3)
- PRA (calculated from the DEM)
- Runout distance as a function of alpha angle (calculated from PRA and Flow-Py)
- Overhead exposure (cell count, $z_{delta}$ or both) (calculated from PRA and Flow-Py)


The first step of the AutoATES classifier is controlled by adjustable thresholds for slope angle, runout distance, overhead exposure and island filter size (Table 1). Using these parameters, the AutoATES model outputs a preliminary, and conservative, layer with the categorical classes (1) simple, (2) challenging, (3) complex and (4) extreme terrain by keeping the maximum value between the 3 input rasters.


Table 1: The recommended input parameters for AutoATES according to Sykes et al. (2023). The encoding describes the name of each parameter in the AutoATES algorithm.

| Input parameter | Class | Range | Encoding |
|---|---|---|---|
| **Slope angle threshold (SAT)** | Simple (1)<br>Challenging (2)<br>Complex (3)<br>Extreme (4) | < 18°<br>18 – 28°<br>28 – 39°<br>> 39° | SAT12=18°<br>SAT23=28°<br>SAT34=39° |
| **Alpha angle threshold (AAT)** | Simple (1)<br>Challenging (2)<br>Complex (3) | < 24°<br>24° – 33°<br>> 33° | AAT12=24°<br>AAT23=33° |
| **Overhead exposure (OE)** | Simple (1)<br>Challenging (2)<br>Complex (3) | < 50<br>50 – 350<br>> 350 | OE12=50<br>OE23=350 |
| **Island filter size (ISL$_{size}$)** | | | 30,000 m$^2$ |

The second step of the AutoATES classifier is to reduce the exposure in certain ATES classes depending on
forest density. The forest density is applied in a secondary step to increase the importance of the forest





density criteria. The forest density layers are divided into four different categories with different thresholds for each forest density input (Table 2).

Table 2: The recommended input parameters for AutoATES according to Sykes et al. (2023). The encoding describes the name of each parameter in the AutoATES algorithm. Only one of the forest inputs can be used at the time, the encoding is therefore identical for all three forest density types.

| Input parameter | Class | Range | Encoding |
|---|---|---|---|
| **Forest density** (Percent canopy cover) | Open<br>Sparse<br>Moderate<br>Dense | 0 – 20%<br>20 – 55%<br>55 – 75%<br>75 – 100% | TREE1=20<br>TREE2=55<br>TREE3=75 |
| **Forest density** (stem density/ha) | Open<br>Sparse<br>Moderate<br>Dense | 0 – 100<br>100 – 250<br>250 – 500<br>> 500 | TREE1=100<br>TREE2=250<br>TREE3=500 |
| **Forest density** (basal area) | Open<br>Sparse<br>Moderate<br>Dense | 0 – 10<br>10 – 20<br>20 – 25<br>> 25 | TREE1=10<br>TREE2=20<br>TREE3=25 |

Once the forest density parameter has been coded into the four classes of forest density (i.e., open, sparse, moderate and dense), as a function of the forest density input parameter used, we mapped these categorical descriptors on to ATES classes (Table 3).

Table 3: Forest criteria applied to the second step of the AutoATES.

| Forest criteria | | Simple (1) | Challenging (2) | Complex (3) | Extreme (4) |
|---|---|---|---|---|---|
| | | | **Initial ATES rating** | | |
| **Open** | PRA & Runout | Simple (1) | Challenging (2) | Complex (3) | Extreme (4) |
| **Sparse** | PRA & Runout | Simple (1) | Simple (1) | Challenging (2) | Complex (3) |
| **Moderate** | PRA | Simple (1) | Simple (1) | Challenging (2) | Complex (3) |
| | Runout | Simple (1) | Simple (1) | Simple (1) | Complex (3) |
| **Dense** | PRA | Simple (1) | Simple (1) | Simple (1) | Challenging (2) |
| | Runout | Simple (1) | Simple (1) | Simple (1) | Complex (3) |

Finally, the island filter size is applied removing clusters smaller than a specified area and incorporating it to the surrounding class. The filter size is not a new addition to the algorithm as it is a part of the v1.0 algorithm, but Sykes et al. (2023) found that a filter size of 30,000 m² (Table 1) was the optimal filter size for all the spatial resolutions tested. The additional step improves the accuracy of challenging (2) and complex (3) terrain, and in some cases in extreme (4) terrain.

## 2.5 AutoATES outputs

The outputs from AutoATES v2.0 have the same spatial resolution as the input. The following outputs are available:

- Continuous PRA
- Flow-Py raw outputs (D'Amboise et al. 2022).
- Preliminary ATES classification of slope angle
- Preliminary ATES classification of runout distance
- Preliminary ATES classification of overhead exposure
- Forest density criteria




- AutoATES v2.0
- AutoATES v2.0 with island size filter

### 2.6 Model assessment

Accuracy, precision, recall, and F1-score are essential metrics for evaluating the performance of a model.
These metrics provide a more detailed assessment, accounting for class imbalance and varying prediction results. They have been widely used in various fields, including avalanche literature (e.g., Keskinen et al., 2022). For a more in-depth understanding of these metrics and their sources, see Liu et al. (2014), who provides a comprehensive review of evaluation metrics for classifiers.

### 3. Results and validation

In order to evaluate the performance of AutoATES v2.0, we use two Canadian benchmark maps made explicitly for Connaught Creek, British Colombia and Bow Summit, Alberta Canada (Figure 3). These are the only locations that have manually mapped maps using the new 5 class ATES model (Statham and Campbell, 2023). The benchmark maps were made by combining individual maps from a panel of three experts, utilizing
methodologies such as Geographic Information Systems (GIS), remote sensing imagery, local knowledge, and field-based investigations. Statham et al. (2023) provide an in-depth description of how the benchmark maps were developed.

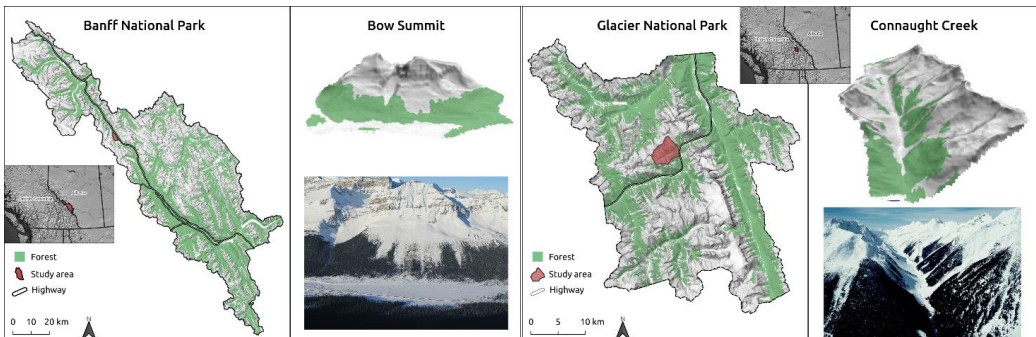

Figure 3: Two areas where benchmark maps for the updated ATES are available is in Glacier and Banff National Park. An overview of the greater area with the study areas in 3D view and overview photo.

### 3.1. Model accuracy

There is no true validation dataset for AutoATES due to differences in scale between automated and manual
methods, but we believe the new benchmark maps made by Sykes et al. (2023) provides the best spatial validation maps to date. In figure 4, we visualize the differences between AutoATES v1.0, v2.0 and the ATES benchmark maps for Connaught Creek and Bow Summit.


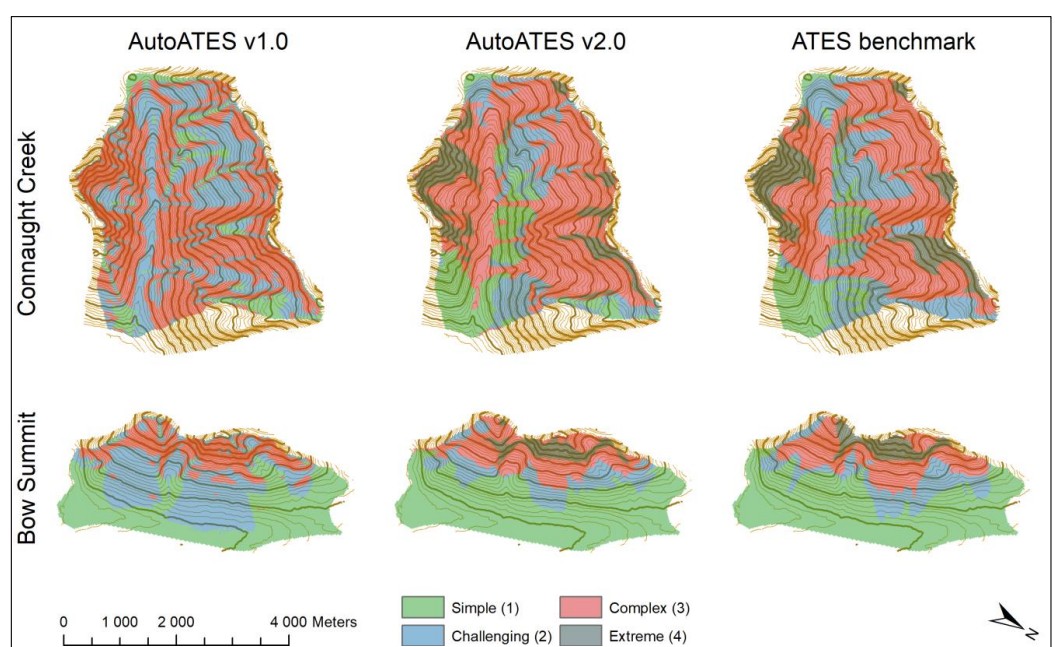

Figure 4: A visual comparison between AutoATES v1.0, v2.0 and the ATES benchmark maps for Connaught Creek and Bow Summit using the European ATES color scheme (Statham et al., 2023). AutoATES v1.0 does not use the extreme (4) class.

We use a confusion matrix for each study area to compare the ATES benchmark, which serves as the ground truth, against the results generated by the AutoATES v2.0 model (Table 4). The confusion matrices enable us to evaluate the performance of the AutoATES v2.0 model by calculating various metrics, such as accuracy, precision, recall, and F1-score. For Bow Summit, the algorithm performs really well for simple terrain with 91.97% accuracy, but the accuracy for challenging terrain is much lower at 65.34%. Complex and extreme terrain is closer to the average with an accuracy of 78.70% and 78.97% respectively (Table 4). The accuracy distribution between the four classes is slightly different for Connaught Creek. The v2.0 model performs the worst in simple terrain with an accuracy of 63.31%. Challenging terrain has an accuracy of 71.0%, complex has an accuracy of 78.0% and extreme terrain has an accuracy of 82.94% (Table 4).

Table 4: A confusion matrix is used to compare the ATES benchmark maps with AutoATES v2.0. Bow Summit is presented above, while Connaught Creek is presented below. The accuracy of each terrain class is marked out with grey shading (area or percent of pixels correctly identified).

| Bow Summit | | AutoATES v2.0 | | | |
|---|---|---|---|---|---|
| | | Simple (1) | Challenging (2) | Complex (3) | Extreme (4) |
| | Simple (1) | 4,527,848 $m_2$ (91.97%) | 140,608 $m_2$ (10.78%) | 16,900 $m_2$ (1.01%) | 0 $m_2$ (0.00%) |
| ATES | Challenging (2) | 391,404 $m_2$ (7.95%) | 852,436 $m_2$ (65.34%) | 179,816 $m_2$ (10.75%) | 0 $m_2$ (0.00%) |
| benchmark | Complex (3) | 4,056 $m_2$ (0.08%) | 310,960 $m_2$ (23.83%) | 1,316,172 $m_2$ (78.70%) | 110,188 $m_2$ (21.03%) |
| | Extreme (4) | 0 $m_2$ (0.00%) | 676 $m_2$ (0.05%) | 159,536 $m_2$ (9.54%) | 413,712 $m_2$ (78.97%) |
| Connaught Creek | | | | | |
| | Simple (1) | 1,364,844 $m_2$ (63.31%) | 263,640 $m_2$ (10.64%) | 76,388 $m_2$ (1.03%) | 0 $m_2$ (0.00%) |
| ATES | Challenging (2) | 683,436 $m_2$ (31.30%) | 1,757,600 $m_2$ (70.96%) | 884,208 $m_2$ (11.92%) | 676 $m_2$ (0.05%) |
| benchmark | Complex (3) | 102,752 $m_2$ (4.77%) | 449,540 $m_2$ (18.15%) | 5,787,236 $m_2$ (78.00%) | 237,276 $m_2$ (17.01%) |
| | Extreme (4) | 4732 $m_2$ (0.22%) | 6084 $m_2$ (0.25%) | 671,944 $m_2$ (9.06%) | 1,156,636 $m_2$ (82.94%) |





A visual presentation of the differences between the two models is shown in Figure 5 where a comparison shows how the models perform compared to the benchmark map for each ATES class, for Bow Summit and Connaught Creek. The bar sections show the absolute accuracy, which is the percentage of pixels that is identical between the benchmark and the automated map. In Bow Summit the v2.0 algorithm has improved challenging terrain a lot with a cost of a small reduction in accuracy of complex terrain. In Connaught Creek, the v2.0 algorithm has improved in all terrain classes, but the improvement is especially clear for simple and challenging terrain.

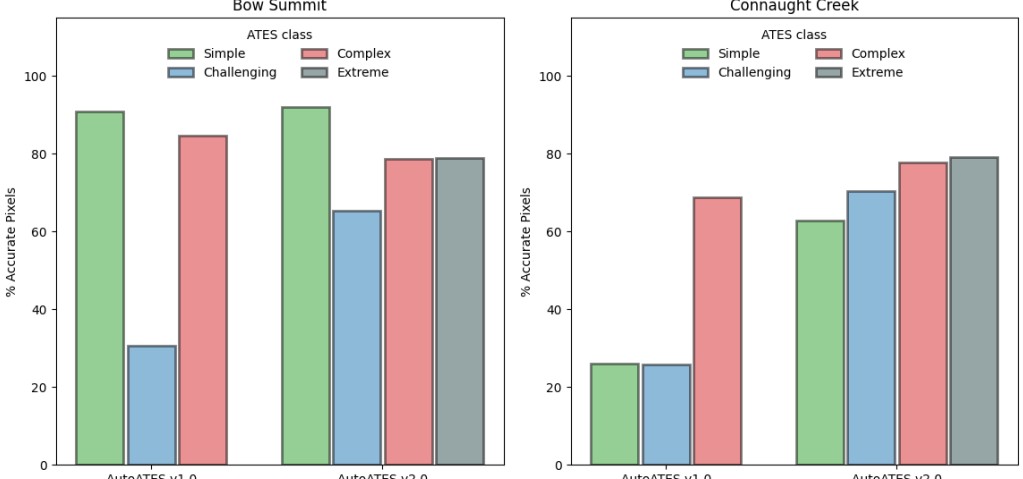

Figure 5: The figure shows how the new AutoATES v2.0 model performs compared to the benchmark maps for Bow Summit and Connaught Creek. The figure uses the European ATES color scheme (Statham et al., 2023). The bar sections show the absolute accuracy, which is the percentage of pixels that is identical between the benchmark and the automated map.

### 3.2 Ablation study

The performance of the AutoATES v2.0 model is a dramatic improvement as compared to AutoATES v1.0. The transition from v1.0 to v2.0 has been marked by numerous internal iterations, featuring improvements such as an optimized PRA algorithm accounting for forest data, incorporating the Flow-Py runout model, considering forest data in the final terrain class algorithm, and more. To fully understand the underlying factors behind the improvements of AutoATES v2.0, it is crucial to examine each of the components that have been modified, which will help clarify how each modification contributes to the overall performance of the algorithm.

To do this, we utilize the concept of an ablation study which is a common method used to evaluate the importance or contribution of individual components within a system, model, or algorithm. It is a type of sensitivity analysis that aims to understand the impact of removing or *ablating* specific components on the overall performance or output of the system. Ablation studies are commonly employed in machine learning, computational neuroscience, and other scientific disciplines to analyze and understand the roles and relationships of different elements in a complex system (Meyes et al., 2019).

The general procedure for an ablation study involves the following steps:

1. Train or develop the full model or system with all its components and parameters intact, and measure its performance on a given task or dataset.





2.  Systematically remove or disable one component or parameter at a time, keeping the rest of the model unchanged.
3.  Measure the performance of the modified model without the removed component or parameter.
4.  Compare the performance of the modified model to the performance of the original, complete model.
5.  Repeat steps 2-4 for each component or parameter of interest.

For AutoATES v2.0, we have identified six components of the algorithm that have been developed since the v1.0. Using the concepts of an ablation study approach, we have calculated the precision, recall and F1-score by removing different components of the algorithm (Table 5). The reference model is the final AutoATES v2.0. The lower F1-score a model has compared to the reference, the more important is the component that has been removed. In Bow Summit, the most important component is the inclusion of forest data in the PRA algorithm (dev4). In Connaught Creek, the most important factor is the post-forest-classification (dev6). In general, all new components in AutoATES v2.0 improve the model by several percents, except the inclusion of AAT3 (dev2), which only improves by 0.08-0.14% for the two study areas.

Table 5: The results from the ablation study where different components are removed to measure the effect for Bow Summit.

| | Version | Component removed | Pixel accuracy | Precision | Recall | F1-score | F1-score change |
|---|---|---|---|---|---|---|---|
| **Bow Summit** | v1.0* | | 67.40% | 68.75% | 66.07% | 64.06% | -13.24 % |
| | dev1* | SAT34 threshold | 87.63% | 78.74% | 76.05% | 81.81% | 4.51 % |
| | dev2 | AAT3 threshold | 84.20% | 82.82% | 80.97% | 77.16% | -0.14 % |
| | dev3 | Forest data from PRA v1.0 | 78.40% | 78.6% | 75.90% | 70.21% | -7.09 % |
| | dev4 | Forest data from PRA v2.0 | 76.80% | 71.29% | 70.61% | 68.03% | -9.27 % |
| | dev5 | Flow-Py (back to TauDEM) | 79.10% | 69.82% | 68.99% | 72.66% | -4.64 % |
| | dev6 | Post-forest-classification | 80.30% | 73.38% | 72.12% | 75.49% | -1.81 % |
| | v2.0 | Reference | 84.40% | 75.74% | 76.19% | 77.30% | 0.00 % |

| | Version | Component removed | Pixel accuracy | Precision | Recall | F1-score | F1-score change |
|---|---|---|---|---|---|---|---|
| **Connaught Creek** | v1.0* | | 49.44% | 40.21% | 38.70% | 38.70% | -32.68 % |
| | dev1* | SAT34 threshold | 80.20% | 72.43% | 74.73% | 72.79% | 1.41 % |
| | dev2 | AAT3 threshold | 74.70% | 73.65% | 70.89% | 71.30% | -0.08 % |
| | dev3 | Forest data from PRA v1.0 | 71.80% | 71.23% | 64.12% | 66.71% | -4.67 % |
| | dev4 | Forest data from PRA v2.0 | 72.70% | 73.33% | 64.68% | 67.73% | -3.65 % |
| | dev5 | Flow-Py (back to TauDEM) | 65.50% | 66.78% | 67.55% | 65.87% | -5.51 % |
| | dev6 | Post-forest-classification | 59.90% | 56.40% | 48.20% | 48.30% | -23.08 % |
| | v2.0 | Reference | 74.90% | 73.80% | 70.94% | 71.38% | 0.00 % |

*\* AutoATES v1.0 and dev1 uses the old ATES v1.0 framework with three terrain classes, which could lead to higher F1-scores. See section 4.1.1 for an in-depth discussion.*

## 4. Discussion

One of the primary challenges when developing AutoATES v2.0 has been to create a robust process for validating the output. Initial attempts by Larsen et al., (2020) compared AutoATES v1.0 to available linear and spatial ATES ratings in Norway, but the validity of these layers was uncertain, given that multiple experts generated them, over a period of years, with limited review.

In contrast, the approach by Sykes et al., (2023), attempts to address these deficiencies, and create benchmark maps for two regions in Canada. Their approach, which used three human ATES mappers who independently mapped each study area, and then created benchmark maps based on their individual output through a detailed discussion of the terrain characteristics, is a more comprehensive methodology to address





this issue. For the purpose of our analysis, we consider these benchmark ATES maps as the standard to which we will measure any AutoATES models to.


When conducting our consensus matrices, we combine non-avalanche and simple terrain to make a 4-class validation dataset to be used against the AutoATES v2.0. We have chosen to not include a non-avalanche terrain class due to the challenges of defining non-avalanche terrain using automated methods.

While the benchmark maps provide the best available validation dataset there are still fundamental differences in how human mappers create ATES maps versus AutoATES. The scale of analysis for human mappers is generally focused on terrain features, classifying an entire ridgeline, bowl, or gulley as a single unit of analysis. In contrast, AutoATES is a raster-based model which operates on a pixel-by-pixel analysis scale. The size of the pixels depends on the DEM data available for a given study area. Variability in DEM
resolution and quality is one of the biggest challenges of applying AutoATES in data sparse regions, like Western Canada. The scale mismatch between human mapped ATES and AutoATES is a persistent difference and an issue that needs to be thoroughly considered with further validation efforts. The optimal scale of use for AutoATES is outside the scope of this current work, but detailed analysis by Sykes et al., (2023) has considered the impact of DEM resolution on AutoATES and notes that there is no real difference in
performance using DEM datasets with a spatial resolution ranging from 5-26 m. We therefore recommend that the spatial resolution of the DEM and forest data is between 5 to 30 meters.

### 4.1 Model performance

We investigate the performance of the AutoATES v2.0 algorithm compared to the v1.0 model, both designed
to identify potential release and runout areas. Although the underlying concept remains consistent between the two versions, numerous components have been altered or refined in the latest iteration.

#### 4.1.1 Extreme terrain (dev1)

The first modification to the AutoATES v2.0 model was to include the extreme terrain class from ATES v2.0.
We incorporated the new class by including another slope angle threshold (SAT). We measure the importance of this change by using the results from the ablations study (Table 5, dev1). The result is that the ablated model performs better with regards to F1-score (i.e., 4.51% improvement for Bow Summit, and 1.41% for Connaught Creek) than the reference model. This means that excluding the SAT34 threshold (i.e., complex / extreme threshold) increases the accuracy of the model. However, without it, the algorithm would be using
the old ATES v1.0 classification excluding extreme terrain. This implies that excluding the SAT34 threshold enhances the model's numerical accuracy. Nonetheless, its absence would cause the algorithm to employ the outdated ATES v1.0 classification, which does not account for extreme terrain, and therefore diminishing its value for ATES v2.0.

When working with classification problems, decision boundaries are the borders or thresholds that separate different classes (Lee and Landgrebe, 1993). The complexity of the decision boundaries often depends on the number of classes. When there are fewer classes, the decision boundaries tend to be simpler, as there are fewer regions to separate in the feature space. With simpler decision boundaries, the model may have an easier time making accurate predictions, as there is less chance of overfitting or incorrectly assigning data
points to the wrong class. This could lead to higher precision, recall, and ultimately higher F1 scores. We believe the fewer classes in the ATES v1.0 is the reason why it performs better than the ATES v2.0 reference model.

#### 4.1.2 Terrain traps (dev2)

To improve the algorithms' ability to identify severe terrain traps such as depressions and gullies, another alpha angle threshold (AAT) was added to be included in complex terrain. The previous model only had AAT thresholds which defaulted terrain into simple and challenging terrain. The extra component was added in the early stages of the development of AutoATES v2.0. The ablation analysis show that this change has a very



little effect on the overall performance of the model (Table 5, dev2). (i.e., 0.14% decrease for Bow Summit,
and 0.08% for Connaught Creek)

### 4.1.3 Forest data in PRA (dev3 and dev4)
Forest density is considered to be one of the most important parameters for ATES classification. In the original
PRA v1.0 from Veitinger et al. (2016) it was not possible to include forest density as one of the inputs. The
modified PRA v2.0 used in the AutoATES v2.0 algorithm builds on the work from Sharp (2018).

The PRA was initially developed and optimized for a 2m DEM, while we utilize a 10m DEM as default. If
roughness was calculated using a 10m DEM, it would measure the roughness at basin scale, instead of the
roughness at the slope scale (Blöschl, 1999; Blöschl and Sivapalan, 1995). The roughness is also dependent
of a snow depth value which is impossible to define without assessing the snowpack properties at a given
time. We do not consider that there is value in running AutoATES v2.0 using high resolution DEMs (< 5 meter).
Sykes et al., (2023) further illustrates the impact of DEM scale on ATES mapping. We have therefore chosen
to remove the roughness parameter from our version of the PRA model.

When comparing the importance of PRA v1.0 (dev3) and PRA v2.0 (dev4) to the reference model, we see that
the forest density into PRA is among one of the most important components (Table 5, dev3-4) (i.e., 7.09-
9.27% decrease for Bow Summit, and 3.65-4.67% for Connaught Creek. Comparing the results between PRA
v1.0 and PRA v2.0, we can measure the difference between the two models without forest input. We found
that the PRA v1.0 performed better than v2.0 in Bow Summit, but the opposite is the case in Connaught
Creek. However, given that Larsen et al. (2020) did not adapt the PRA v1.0 algorithm according to the
recommendations of Veitinger et al. (2016), we believe the changes are conceptually still important even
though there are no substantial differences between the two in the ablation validation.

### 4.1.4 Flow-Py (dev5)
The previous iteration of AutoATES had some severe issues with the runout simulation of avalanches where
avalanches where simulated using a flow model for water. The Flow-Py simulation works in a similar fashion
where the flow is limited by an alpha angle threshold, but the flow model has been changed to give more
realistic outputs in terms of snow avalanches. Some other advantages with the Flow-Py simulation suite are
that there are additional outputs such as cell count and $z_{delta}$ which makes it possible to account for the
exposure of multiple overlapping paths and avalanche paths with high kinetic energy. When we compare the
Flow-Py outputs compared to the TauDEM, we see a substantial improvement when using the Flow-Py
outputs (Table 5, dev5) (i.e., 4.64% decrease for Bow Summit, and 5.51% for Connaught Creek).

### 4.1.5 Post-forest-classification (dev6)
Even though the inclusion of forest density in the PRA algorithm improved the performance of AutoATES, we
found the need to reclassify sections that obviously where densely forested and resulted in a higher ATES
rating than needed. To improve this, we added a post-forest-classification criteria. This was really efficient
for Connaught Creek, but less efficient for Bow Summit (Table 5, dev6) (i.e., 1.81% decrease for Bow, and
23.08% for Connaught Creek). The forest impact of dev6 is minimal at Bow Summit, but really important for
Connaught Creek. We don't know why this is, but one hypothesis is that there is more steep forested terrain
in Connaught Creek, and the algorithm therefore relies more on the post-forest-classification. Connaught
Creek also has more large runouts and overhead hazard that rely on the post-forest-classification.

In the future, we hope to be less reliant on the post-forest-classification criteria by optimizing the forest
detrainment module in Flow-Py. This module of Flow-Py makes it possible to reduce the runout length in
areas with dense forest.

### 4.1.6 Discrepancies



The discrepancy in accuracy scores between the two study areas is mainly attributed to the complex terrain of Connaught Creek with many smaller topographical features and the limitations of the VRI forest data resolution in capturing local forest characteristics (Sykes et al., 2023). This issue significantly affects the assessment of overhead hazards and boundaries delineation between ATES classes, with challenging (2) terrain showing the lowest accuracy and high rates of underprediction errors. Sykes et al. (2023) provides an extended discussion of the differences between the two study sites.


### 4.3 Application

While it is possible to run the presented version of AutoATES v2.0 without making any changes, we recommend a workflow where the optimal parameters are first identified. The suggested parameters in this paper are valid for the two test areas in Western Canada, so when applying AutoATES v2.0, there will likely 575 need to be a reevaluate the parameters for the area being mapped. Blindly applying the parameters presented in this document to other regions without site specific calibration risks inaccurate ATES mapping, and potential catastrophic outcomes. Users should apply at their own risk. We therefore urge all future users of our code to conduct, and document, their local validation before proceeding with the generation of ATES maps, especially when the intended target is the general public.


Begin with a relevant test area which should include a variety of terrain and all terrain classes. We recommend a workflow where the PRA model and Flow-Py is processed independent of the AutoATES classifier. The output from PRA and Flow-Py is easier to validate by local experts compared to the AutoATES output. It is more intuitive as avalanche experts have more tangible experience with identifying start and 585 runout zones. In our experience, we complete approximately 1-3 iterations of PRA and Flow-Py before moving on to the AutoATES classifier. In general, we have experienced that the 'c' parameter in the Cauchy function for slope angle combined with the max alpha angle for Flow-Py are the most effective for customizing the output. We also recommend fine-tuning all parameters in the Cauchy function for PRA when using new forest density data.


When these steps are done in advance, our experience is that the output of the AutoATES classifier tends to be much more accurate. The final AutoATES could then be shared among local experts which provides further feedback. Changes could then be made to the AutoATES classifier parameters and improved during an iterative process. When the final input parameters are set, they could be used to generate larger areas. A 595 description of the input parameters used should be shared as meta-data with the resulting spatial maps.

### 4.4 Limitations

Despite the notable improvements of the AutoATES v2.0 model, there are still some limitations that should be acknowledged.


- Flow-Py is computationally heavy, which may present challenges when processing large datasets or applying the model in real-time applications. This could potentially limit the scalability and accessibility of the model for certain use cases and users with limited computational resources.
- Determining the optimal input parameters for the AutoATES model is important to get the best 605 performance possible. The suitability of these parameters across different snow climates and terrain types remains an open question. Further research and validation are needed to ensure that the chosen parameters provide accurate and reliable results in various contexts. Users should not blindly adopt the input parameters stated in this paper.
- The model does not account for changes in vegetation over time such as natural events like landslides 610 or forest fires. Therefore, it is important to update the ATES mapping periodically to account for major changes in the landscape.



Addressing these limitations in future work could enhance the performance, applicability, and reliability of the AutoATES model, ensuring its effectiveness across a wide range climates and terrain characteristics.


**5. Conclusion**

In conclusion, the development of AutoATES v2.0 has focused on creating a more robust and accurate algorithm for mapping avalanche terrain into ATES ratings by incorporating new components to improve the algorithm. This has been achieved by integrating new components that enhance the algorithm's

performance, including the addition of an extreme terrain class, improved PRA with support for multiple forest density types, Flow-Py, and a post-forest-classification criteria. Moreover, a significant portion of the code has been rewritten to increase efficiency and eliminate dependency on proprietary software.

However, limitations related to the determination of optimal input parameters for different regions and

climates need to be considered for future model development. By addressing these limitations and continuing to refine the model through iterative testing and expert feedback, AutoATES v2.0 can serve as a valuable tool for avalanche risk assessment and decision-making in a wide range of snow climates and terrain types. Ultimately, our goal is for AutoATES v2.0 to enable efficient, large-scale, and potentially global ATES mapping in a standardized manner.


**6. Code and data availability**

To reproduce the results from this study, please find the AutoATES v2.0 algorithm and validation data from the ablation study in the OSF repository. For future application of AutoATES v2.0, a GitHub repository will be maintained with future iterations of the algorithm available (Toft et al. 2023).


**7. Author contribution**

HT was the developer of the first version of automated ATES. The new versions improvements have been led by HT with significant contributions from JS and AS. The ablation study has been carried out by HT with inputs from JS. HT prepared the final manuscript with JH and AH as editors. All co-authors contributed to the final

manuscript.

**8. Competing interests**

The authors declare that they have no conflict of interest.

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
