# Peer review of "AutoATES v2.0: Automated avalanche terrain exposure scale mapping"

_Natural Hazards and Earth System Sciences, 2023_

## Author Comment (AC1)

**Response to reviewer comments**

Toft, H. B., Sykes, J., Schauer, A., Hendrikx, J., and Hetland, A.: AutoATES v2.0: Automated avalanche terrain exposure scale mapping, Nat. Hazards Earth Syst. Sci. Discuss. [preprint], https://doi.org/10.5194/nhess-2023-114, in review, 2023.

**Review 1 - Scott Thumlert**

**Overall**

The paper presents an improved automated avalanche terrain classification model (AutoATES v2.0) which is a valuable contribution in the specialized field of describing and mapping the severity of avalanche terrain. Many novel and appropriate scientific techniques have been applied in the development and assessment of the model which are worthy of publication. However, I recommend revisions be completed before publication. I hope that this review helps to improve the overall quality of this excellent manuscript.

**General comments:**

The writing in the paper suffers greatly from what can be described as expert familiarity. That is, the authors are obviously experts with and extremely familiar with the subject matter, which has led to a natural assumption that the readers have similar understanding. I recommend the authors have the manuscript reviewed by a non-expert to identify areas where the manuscript can be clarified.

**The manuscript will be reviewed by a non snow and avalanche expert, but someone familiar with the NHESS readership, to improve areas that should be described more in detail.**

The manuscript would benefit greatly from fixing the many comma splices found throughout. I have identified many in the technical corrections section, however I recommend a thorough review.

**Thank you, one of our native English-speaking co-authors will do a thorough review of all comma splices, after review by the above non snow and avalanche expert.**

Lines 27-28: Confirm that the two significant digit accuracy is valid and useful, i.e. does the level of accuracy increase the reader's understanding? Further, the authors use both one and two significant digits in similar results. Please explain why varying significance is valid or be consistent across the manuscript.

**We will remove the two digits and only present integer values as the two-digit accuracy is not adding any information to the reader.**

Abstract: The abstract requires significant improvement. Overall, it assumes the reader is familiar with the Avalanche Terrain Exposure Scale (ATES) and AutoATES v1.0, and thus does not adequately introduce the overall topic of rating the severity of avalanche terrain. Suggest that the distinction between linear and spatial ATES ratings be introduced. Suggest introducing AutoATES v1.0. I'd also recommend expanding on the improvements from AutoATES v1.0 by explaining the forest data improvements to the PRA algorithm, the use of Flow-Py, and the post forest classification.

**We will rewrite the abstract to include the topics suggested above. Thank you for highlighting our familiarity with the topic, and lack of introduction to key topics and concepts in the abstract.**

Lines 41 and 42: Assessing avalanche risk indeed is a complicated task for backcountry recreationalists. Lines 41 and 42 imply that spatial and temporal variability are the reasons for the complexity, however these factors certainly contribute to the complexity, but are not the sole reasons. Suggest changing the wording of this sentence.

**We will change the wording of this sentence as suggested.**

Lines 43-44: The sentence when interpreted with the preceding two implies that backcountry recreationalists make detailed assessments of instabilities at a regional scale (i.e. > 104 km2). Backcountry travelers typically travel on the mountain or drainage scale (i.e. > 102 km2), therefore they are unlikely to make detailed assessments at the larger regional scale. Regional scale assessments are often done by public forecasters. Further, avalanche risk when travelling in avalanche terrain is managed by more than just assessments of weather, snowpack, and signs of instability. Other risk management techniques include terrain selection, travel techniques to reduce exposure to hazard (e.g. not stopping on an avalanche slope), the use of airbags, and the use of rescue equipment (e.g. transceivers, probes, shovels). Suggest revising the sentence.

**We will revise this sentence as suggested.**

Lines 37 and 38: Please explain how the 90% triggering statistic is relevant to the topic.

**We will change the citation with a more recent one from Techel et al. (2015) and expand in the lines of:**

**"More than 80% of fatal avalanche accidents are related to recreational activity and triggered by the victim or someone in their party (Techel et al., 2015). This means that avalanche accidents are not random, but rather a result of less-than-optimal decisions. Strengthening people's ability to make better decisions by raising awareness, providing information and education is important and may ultimately save lives."**

**Techel, F., Zweifel, B., and Winkler, K. (2015). Analysis of avalanche risk factors in backcountry terrain based on usage frequency and accident data in Switzerland, Nat. Hazards Earth Syst. Sci., 15, p. 1985–1997, https://doi.org/10.5194/nhess-15-1985-2015**

The introduction needs to describe the distinction between the original use of ATES to rate linear routes (i.e. backcountry trips) and the development of spatial ATES ratings. It is critical for readers to understand that ATES was developed initially to describe the severity of avalanche terrain one would be exposed to when travelling on linear routes through mountain terrain. The discussion and/or the introduction sections would benefit from a discussion of the implications of using ATES spatially which is a deviation from its original intention.

**We will expand on the distinction between linear and spatial ATES and include a section in the introduction and discussion on how our application of ATES deviates from the original purpose.**

Lines 212 and 213: Should this sentence refer to "stems per hectare" rather than "stem density per hectare"? Stems per hectare is commonly used to describe forests. If stem density per hectare is correct,

please define the unit. Also, it is not possible to have infinite stems / hectare as this implies infinitely small stem diameter.

**This is a spelling error, and we will change to stems per hectare. We will also change the sentence stating it is possible to have values from 0 to infinity. The idea was that there is no maximum in value, but of course in reality at some point, there can't be any more stems within a hectare. We will change this wording into something like "a value ranging from 0 to a few thousands (depending on minimum stem diameter)".**

Suggest referring to the new 5-class ATES rating system recently published by Statham and Campbell (2023) as "ATES v2.0". Then refer to the AutoATES algorithms as AutoATES v1.0 and AutoATES v2.0 throughout. For example, the clarity of the sentence on Line 401 could be improved if v2.0 was replaced with AutoATES v2.0.

**We will  make changes to the manuscript so we always reference the different ratings as ATES v1.0/v2.0 and AutoATES v1.0/v2.0.**

The study evaluates the output from the AutoATES algorithms against two benchmark maps: 1) Bow Summit, and 2) Connaught Creek. An improvement to this study and future studies would be to expand the amount of terrain in the benchmarks to allow a more robust evaluation of the output. For example, avalanche terrain in the coastal mountain range differs in character to the interior and Rocky Mountain ranges in Canada, and it would be useful to see how the model performs in the coast. Also, terrain in different countries would be valuable to evaluate (e.g. the Alps, Japan, Sierras). Please speculate on how the model would perform in different snow and avalanche climates.

**We agree that expanding the validation dataset to include areas with maritime snow climates and a wider variety of terrain types is a critical next step, but is outside the scope of this current paper. Future research efforts will aim to include additional validation areas and consider how data availability in different countries could impact performance of AutoATES. This limitation and suggestions for how the model could adapted for different study areas are included in the companion paper (Sykes et al., 2024).**

**Sykes, J., Toft, H., Haegeli, P., and Statham, G.: Automated Avalanche Terrain Exposure Scale (ATES) mapping – Local validation and optimization in Western Canada, Nat. Hazards Earth Syst. Sci. Discuss. [preprint], https://doi.org/10.5194/nhess-2023-112, in review, 2023.**

It does not appear as though the authors include Class 0 - Non-Avalanche terrain as an output of the AutoATES v2.0 algorithm. Please provide a robust explanation of why this was excluded and the Class 4 - Extreme terrain was included. Class 0 - Non-Avalanche terrain is a critically important classification. I strongly suggest the authors revise the algorithm and manuscript to include this classification level which would increase the value of the scientific and practical contribution.

**Due to the limited sample size of mapped class 0 terrain in the validation data sets that we used to develop autoATESv2.0, we do not feel that there has been sufficient research on this topic to warrant publication at this time. AutoATES is a promising tool for estimating areas with no exposure to avalanche terrain, however there is significant liability associated with deeming an area safe from avalanche hazard. Further development of the autoATESv2.0 model and consultation with avalanche community stakeholders is necessary before delving into automated mapping of class 0 terrain. We**

**will add a discussion to the manuscript regarding our rationale for not including class 0 terrain, and the potential for future developments along these lines.**

Lines 89 to 90: The term zoning is often used for avalanche hazard zoning where the zones specify requirements for land use (e.g. building not permitted in red zone). Suggest using "avalanche terrain classification" here. Also, I'd suggest making very clear distinctions between hazard zoning or mapping and terrain classification / mapping. Avalanche terrain is classified mostly independent of hazard and is much different than hazard zoning. Consider removing any reference to hazard zoning from the manuscript for clarity.

**Thank you for identifying this, we will remove all references to hazard zoning to avoid any confusion.**

Line 156: Recommend ending the introduction with the proposed solution that solves the identified knowledge gaps and improves on the identified shortcomings of AutoATES v1.0.

**We will add a paragraph that presents the goals of this paper in terms of improving the identified knowledge gaps and improvements specifically from AutoATES v1.0**

Line 158: Section 2. Model Motivation appears to include descriptions of the model methodology and development. Consider a more appropriate title for this section - perhaps "2. Model Methodology" or "2. Model Development".

**We will change the title for this section to "2. Model development"**

Lines 195 to 204: Forest canopy also impedes wind transport of snow reducing the formation of wind slabs. Suggest adding this to the description of the effect of forest canopy on avalanche formation.

**We will add this sentence to lines 195 to 204.**

Line 307: Please clarify if the PRA is calculated from the DEM only or is forest density considered as well.

**This is an error. The PRA is calculated from the DEM and forest density data. We will add this correction in the revised manuscript.**

**Technical corrections**

Many technical corrections were found. I recommend the authors thoroughly review subsequent versions of the manuscript prior to submission.

Lines 22-24 - clarify the sentence. It is not clear what is meant by "open-source software" in relation to addressing the significant drawbacks of AutoATES v1.0. A comma in Line 24 after "overhead exposure" would help.

**We will add a comma as suggested. We will add the context that we are making the v2.0 algorithm available using open-source software in contrast to v1.0**

Line 30: Perhaps a missing word at the end of the sentence? Should this read "decision-making".

**You are correct, we will add the missing word in the revised manuscript.**

Lines 24 and 31: Suggest defining ATES before using the acronym or not using the acronym in the abstract at all and writing out the full "Avalanche Terrain Exposure Scale".

**We will define the ATES acronym before using it in the revised manuscript.**

Line 42: Suggest replacing "snow" with "avalanche hazard".

**This sentence will be rewritten as part of responding to one of the specific comments above.**

Line 52: ATES is an ordinal scale. Ordinal scales are not typically described as quantitative measurements. Replace "quantifies" with something like "classifies".

**We will replace the word with "classifies".**

Line 44: Delete ",i.e.,"

**This will be deleted.**

Line 48: replace the "," before "however" with a semicolon, OR delete the "," after "however".

**Comma will be replaced with a semicolon.**

Line 50: ATES communicates the severity of avalanche terrain and not the risk. Avalanche risk is often described as a function of expected avalanche size, likelihood, exposure, and vulnerability; therefore I'd suggest deleting "risks" from this sentence. Parks Canada originally used the system to describe the exposure to avalanche terrain one would experience on a particular backcountry trip.

**We will remove the word risks and incorporate exposure to avalanche terrain into the sentence.**

Line 51 and 52: Appears to be a missing word between scheme and worldwide. Consider: "ATES is a terrain classification system commonly used worldwide to describe avalanche terrain using easy-to-understand ratings: Simple (1), Challenging (2), and Complex (3). Also, suggest capitalizing Simple (1), Challenging (2), Complex (3), and Extreme (4) here and elsewhere for clarity that these are output variables from the model.

**We will add the suggested sentence, thank you.**

Lines 55 to 57: Rewrite sentence to clarify that the new ATES system is called ATES v2.0. Consider: "Recently, the ATES classification system has been updated to include two new additional ratings: 1) Non-avalanche terrain (0), and 2) Extreme (4). The updated system complements the existing system and is now referred to as ATES v2.0 (Statham and Campbell, 2023)." Suggest using consistent ATES terms with capital first letter and paratheses containing the level.

**We will add the suggested sentence, thank you.**

Lines 59 to 70: The connection between avalanche hazard mapping (i.e. magnitude x return period) and terrain classification is vague. Consider removing the hazard mapping description unless the authors feel it adds value to the topic of avalanche terrain classification.

**We will remove all references to hazard mapping as suggested.**

Line 61: Suggest replacing "i.e.," with "e.g.,". The authors are providing an example and not further describing the definition of specific return periods. That is, hazard mapping is often done for other return periods depending on the context.

**This will be replaced with "e.g." as suggested.**

Line 92: Consider replacing ", i.e., terrain traps and forest density," with (e.g. terrain traps, forest density). These are examples of other factors but not a comprehensive list.

**This will be replaced with "e.g." as suggested.**

Line 95: Comma splice between width and accounting.

**This will be fixed.**

Line 112: Add comma after "et al.". Here and throughout.

**This will be fixed.**

Line 113: Missing parentheses around "2016".

**This will be fixed.**

Line 123: Replace "is" with "are". Common error throughout the manuscript.

**This will be fixed.**

Line 133: Remove the "s" in the word "defines".

**This will be fixed.**

Line 162: Comma splice after the Sykes reference. Add a conjunction such as "which limits".

**This will be fixed.**

Line 163: Comma splices. Consider: "For AutoATES v2.0 to be a viable option for largescale ATES classification, the model performance should be at least as accurate as manual mapping."

**This sentence will be replaced with the suggested one, thank you.**

Line 169: Suggest re-writing this sentence. It is not clear what is meant by "integrating the parameter into track and deposition area".

**This sentence will be rewritten.**

Lines 166 to 175: Regarding the model description. Doesn't the model also include post-forest-classification? Consider clearly describing the main model additions with a description of each. This will help add clarity to the model assessment and discussion sections.

**You are correct, we will add a description for the post-forest-classification in the revised manuscript.**

Line 181: Delete "Sykes" from the reference.

**This will be deleted.**

Lines 191 and 192: Confirm that the algorithm uses all of these forest data types at once OR if the algorithm uses only one of those forest data types. If the algorithm uses one of the forest types, then replace the word "and" in line 192 with "or". Same comment for Line 194 in the parameter list.

**We will replace "and" with "or" as we only use one of the forest data types at once.**

Line 225: replace "is" with "are".

**This will be fixed.**

Line 237: add comma after "et al.". Here and elsewhere in the manuscript.

**This will be fixed.**

Line 239: delete the "s" on the word "values".

**This will be fixed.**

Line 241: Suggest using semicolons between the main list items: "where u(x) is the Cauchy membership value; x is an input variable; and a, b, and c are parameters ...".

**This will be fixed.**

Line 255: Comma splice. Re-word the final sentence in the caption.

**This will be fixed.**

Line 256: Replace "is" with "are".

**This will be fixed.**

Lines 363: Suggest referring to the new ATES system as "ATES v2.0 (Statham and Campbell, 2023)". Here and throughout for clarity.

**Agreed, this will be fixed.**

Line 370: Figure 3 – the inset labels are ineligible which makes placing the study areas difficult for readers who are unfamiliar with the locations.

**We will update the figure in the revised manuscript to be more readable.**

Line 381: The reference should be (Statham and Campbell, 2023) because there are only two authors. Check all references and citations.

**We will check all the references for this error in the revised manuscript.**

Lines 388 to 392: Use a consistent level of significant digits for the model accuracy values OR explain why different levels of accuracy are relevant.

**We will change to whole integer values.**

Table 4: The headings for SIMPLE (1) and CHALLENGING (2) look to be misaligned.

**You are correct, this will be fixed.**

Line 400: Replace "is" with "are". Please check the entire manuscript as this error repeats.

**This will be fixed.**

Line 408: Should the reference be "(Statham and Campbell, 2023)".

**This will be fixed.**

Line 413: Avoid words like "dramatic" when describing scientific findings.

**We agree, the sentence will be rewritten to avoid overly emotive language.**

Line 445: Define AAT3.

**This should have been AAT23, and it's defined in Table 1. We will fix this.**

Table 5: Suggest defining SAT34 and AAT3 in the table caption so that the table can stand alone from the text.

**We will add a description to the table caption.**

Lines 456 and 457: Comma splice. Suggest re-writing this sentence with connectors. For example: Initial attempts by Larsen et al., (2020) compared AutoATES v1.0 to available linear and spatial ATES ratings in Norway, however the validity of these layers was uncertain because they were developed over multiple years by numerous experts with limited review.

**We will change the sentence as suggested.**

Line 459: delete the commas after (2023) and deficiencies.

**This will be fixed.**

Lines 460 - 463: Re-write the sentence please because it runs on. Suggest: "Their approach - which used three experts to map each study area and then create benchmark maps based on their individual output - is a more comprehensive methodology to address this issue."

**We will change the sentence as suggested.**

Line 466: Does the term consensus matrices refer to the confusion matrices? If so, please adjust.

**Should be "confusion matrices", and this will be fixed.**

Line 471: Suggest replacing "human mappers" with something more suitable like "terrain rating experts" or "avalanche experts". If agree, do so here and elsewhere. The change will improve the sentence by reducing the repetitive use of "mappers" and "maps".

**We will replace the term human mappers with terrain rating experts throughout the manuscript.**

Line 475 and 476: suggest replacing the ", like Western Canada" with "(e.g. Western Canada)".

**We will change as suggested.**

Line 484: Change "investigate" to "investigated". Also, fix the comma splice after "model".

**This will be fixed.**

Line 490: Change "measure" to "measured".

**This will be fixed.**

Line 510: Does the model identify other common terrain traps such as cliffs, crevasses, forest? If not, please provide rationale and speculate on the effect on model performance if all common terrain traps were included.

**The addition of another AAT was only made to incorporate depressions and gullies as this was a simple feature to include. Cliffs of a certain size are covered by the extreme SAT34, and smaller features (less than 10-30 m) would not be viable given our DEM´s spatial resolution. Incorporating forest and crevasses terrain traps are specific type that are more complex to model, and we did not attempt this at this stage. We will include a couple of sentences explaining this in the revised manuscript.**

Line 532: Missing parentheses.

**This will be fixed.**

Line 565: VRI is not defined.

**Should be "Vegetation Resources Inventory". We will correct this in the revised manuscript.**

Line 575: Re-write sentence. Consider replacing "reevaluate" with "re-evaluation".

**We will replace the word and rewrite the sentence.**

Lines 577 to 579: Re-write sentence to reduce the comma splices and increase clarity.

**We will rewrite the sentence.**

Lines 635 to 640: Write out the abbreviated names for clarity.

**We will fix this in the revised manuscript.**

**Reviewer 2 – Anonymous**

**Overall**

This paper details the latest advances to the spatial ATES mapping approach, called AutoATES v2.0, to improve an automated terrain classification system for snow avalanche hazards at a regional scale. The authors use two study sites to demonstrate the improvements to the earlier version of AutoATES as well as a reference classification based on manual mapping by experts.

The paper topic is of interest to the natural hazard's community and specifically the snow avalanche community. It is generally well-written; however, the structure of the paper needs to be improved to aid in interpretation, and there is important context missing from sections. The paper is written as a companion article, but this is not made clear enough and probably explains some lack of broader context/background on ATES and spatial ATES. A review from a non-expert would help ensure broader interpretation of the research.

Some significant revision is required before the paper will be ready for publication. Please find general comments below followed by specific comments/technical points.

**General comments**

The paper is generally well-written, but the structure of the paper needs significant re-organization and improvement. Sections in the discussion repeat what is written earlier and, in some cases, provides more detailed information that would aid the reader had it been introduced earlier. For example, the benchmark sites are referenced early but not properly introduced until the Discussion.

**We will introduce the benchmark maps earlier and ensure that we read through the manuscript thoroughly to remove repeating information in the revised manuscript. Reviewer #1 also recommended a non-expert review the final version of the manuscript, which will ensure another level of review to minimize duplication and redundancy.**

The abstract needs to be reformulated to bring better context to the paper. It currently reads more like a description of the algorithm, but as this is a general hazards journal, it should be written so someone can get all the context needed to understand the contents of the paper before diving in. In the first instance, please provide a (brief) introduction of ATES and AutoATES. I believe you intend 'large-scale mapping' to refer to regional or landscape (10 or 100s km2) mapping products, but the term actually conveys a map covering a small geographic extent. 'Regional-scale mapping' may be more appropriate.

**Thank you for the feedback, we received similar feedback from Reviewer #1, and will rewrite the abstract to include more general context to the study. We will include the concept of rating avalanche terrain, introduce ATES and the new version of ATES, and that ATES was traditionally a linear classification which we now apply spatially. Then introduce AutoATES.**

**We will replace with "regional-scale mapping" as suggested.**

The writing feels like the reader should already be an expert on all aspects of ATES and AutoATES development. You do not need to repeat everything covered in the companion article, but context is missing throughout. For example, the forest data layers are a big part of the focus of this paper. There is no detail on what dataset was used, how any uncertainty on the rasterization of those data was accounted for in their use in the PRA tool or the classifier, etc, how users without your forest data product should approach the use of other forest data.

**The companion paper utilized a vector-based forest inventory dataset available in British Columbia (British Columbia Vegetation Resource Inventory), Canada and Banff National Park in Alberta, Canada. The forest polygon data was converted to raster to match the resolution of the input DEM using a nearest neighbor resampling method. We will add these details into the manuscript discussion. Furthermore, future research will aim to document methods to use satellite remote sensing data to generate consistent forest input data worldwide using image classification methods (Sykes et al., 2022).**

**Sykes, J., Haegeli, P., and Bühler, Y.: Automated snow avalanche release area delineation in data-sparse, remote, and forested regions, Nat. Hazards Earth Syst. Sci., 22, 3247–3270, https://doi.org/10.5194/nhess-22-3247-2022, 2022.**

Overall the paper feels like it was written for an audience of expert ATES users who might be trying to run AutoATES v2.0. This should be broadened for a general audience of the journal NHESS. Some sections (like Application) are important for broadening the reach of the paper but currently feel like they were dropped in without connecting to the rest of the paper. For example the statement in Lines 588-589 is a good idea but more specificity is needed on how a user would fine-tuning. What new forest density data are you referring to?

**We will improve the application section in the revised manuscript according to the comments provided by Reviewer #1 and Reviewer #2.**

**By new forest data, we will add the following to provide clarity: "We also recommend fine-tuning all parameters in the Cauchy function for PRA when using other forest density data than what's being validated in this paper."**

**We will also add the sentence: "This could be done by using a local avalanche terrain expert to review the output from each Cauchy membership value and adjust until the output is appropriate." To more specifically explain how a user would do the fine tuning of the PRA".**

The goal of this work is to create a product that is more standardized and globally applicable than what can be created ad-hoc by experts, but the product is tested against a reference from experts interpreting largely the same data (but with field observations and oblique images etc. as well). This approach is justified but take care in how this is expressed. Should the AutoATES replace the expert ATES mapper or is it meant to be a tool for the expert to use when doing ATES mapping in a new location? Or something else? This should be made clear.

**AutoATES is meant to be a stand-alone tool for mapping large-scale areas, but it should first be validated for a smaller area by experts to assess whether there is a need to make some changes to the input parameters. When the user is confident with their maps, the parameters could be used to generate ATES maps for a larger surrounding area with similar snow and terrain properties. We will expand on this in section 4.3 – Application, specifically noting the anticipated scope of inference.**

One main audience for AutoATES is those wishing to classify terrain in data sparse regions. The paper should make clearer how to use AutoATES v2.0 in these data sparse regions. What global forest product is recommended? How do we handle the uncertainty that is created when resampling these data to match the resolution of the DEM used?

**To our knowledge there are no global forest data sets available that have suitable accuracy and resolution for AutoATESv2.0. In each country we have tested AutoATES (Norway, Canada, USA) there has been a considerable testing period to determine the best available forest data and fine tuning of model parameters to work well with local forest data. This is the rationale for providing multiple 'default' settings for the input forest data including stem density, canopy cover, and basal area. The PRA parameters used for each of these are unique and need to be locally tested before large scale application of AutoATESv2.0. We will add a discussion on these points to the manuscript in tandem with the responses to the prior comments about forest data sets.**

There is a general tension in the paper between fine-scale terrain features and broad-scale terrain classification. Recommendations are made for fine-tuning to local conditions (adjusting DEM resolution, forest data, the use of PRA, etc) but this suggest the AutoATES v2 is sensitive to scale and terrain shape.

A section that details (or speculates) on how the performance changes in different kinds of terrain and forested landscapes would aid the application to other regions.

**We identified that the two study areas both represent large alpine mountain regions with very large avalanche paths and well-defined runout zones. Further validation in a variety of terrain types, including forested areas, rolling hills, and other types of climates, are future research goals. We do not wish to speculate on how the performance would change in different kinds of climates.**

The forest density approach is an intuitive improvement over binary forest presence/absence, but there is no demonstration of how these density products (and the thresholds used) transfer to the avalanche hazard. Please refer to other research/literature that helped inform the values used in thresholds. There are several density measures used here. Help the reader to understand your preferred option (pcc, stems etc). Why include all three? These answers may be straight-forward, but they should be made explicit to the reader.

**Our goal in supporting multiple types of forest data is to make the model more flexible to accommodate locally available forest data. A variety of different spatial data products are available to characterize forest density in different regions, including vector-based data sets that are human generated and raster-based data sets that are extrapolated based on measurements from forest inventory plots. The variables we included represent the most meaningful metrics available in the Norwegian and Canadian test sites (canopy cover, stem density, basal area). Future development could help standardize forest classification approaches and help improve consistency in the application of AutoATESv2.0 across international borders. We will expand section 2.3 to provide some more context.**

Relatedly, please clarify why a binary forest product is not an option since this will be a more widely accessible global product over forest density and the density values are collapsed into three encoding classes.

**We explain this in lines 114-118.**

**"In prior automated ATES mapping work, Larsen et al., (2020) used the PRA algorithm of Veitinger et al. (2016) for the AutoATES v1.0 algorithm due to the continuous raster output. While it is possible to include a binary forest parameter in the Veitinger et al. (2016) PRA model the binary nature of the parameter results in coarse output, as the model removes all PRAs when the forest parameter takes the value 1."**

**The simple binary approach only allows us to separate between open terrain and forested terrain. If it is forested terrain, it is not a potential release area. However, this simplified approach does not allow us to incorporate the forest interaction between dense forest (no avalanches can release) and open terrain. In reality, a large percentage of skiing terrain is in this continuum.**

Please specify what forest product you used (only an acronym is provided). From the companion paper I can see it was created from a vector polygon dataset. What was spatial scale used to create the vector dataset and is rasterizing to 10m appropriate based on the scale or is there additional uncertainty generated? The best global forest density product I am aware of is 30m. This uncertainty may be incidental but the process should be documented.

**The companion paper utilized a vector-based forest inventory dataset available in British Columbia (British Columbia Vegetation Resource Inventory), Canada and Banff National Park in Alberta, Canada. This data set is generated by human interpretation of high-resolution aerial imagery, combined with extrapolation from a network for forest inventory plots. The forest polygon data was converted to raster to match the resolution of the input DEM and used a nearest neighbor resampling method. There is inherent inaccuracy involved with converting the vector data to raster, however the impact on the overall map accuracy and output is minimal. We will include some more details on this in section 2.3.3.**

Scale matters for the DEM as well. While a DEM sensitivity test showed little change to the final ATES mapping (Sykes et al., 2024), this seems to be mostly related to increasing the slope window size to only look at broad landscape-scale features rather than fine-scale features. Flow-py modelling will be sensitive to DEM resolution and whether a DTM or DSM is used. If one advantage of Flow-py is the ability to capture terrain traps and confined terrain then a higher-resolution DEM is needed. Can you clarify if these features are captured in the final ATES class, i.e. do you see terrain with fine-scale features like terrain traps reclassed as a result of using Flow-py? Since ATES mapping is more about broad-scale terrain I suggest reviewing the discussion on fine-scale terrain.

**We agree that DEM scale has a large impact on the PRA model and Flow-Py runout simulations. The validation study did not address these questions. Further research is required to better understand how DEM resolution impacts the AutoATES workflow upstream of the ATES classification. In general, Flow-Py performs well in capturing terrain traps due to avalanche runout regardless of DEM resolution. For coarser resolution DEM input data, the Flow-Py simulation tends to spread avalanches out more across the slope due to the higher degrees of smoothing inherent in a lower resolution data set. However, our experience indicated that the wider lateral spreading caused by lower resolution DEM data still aligns to known avalanche path boundaries as identified by forest trim lines. These results are discussed in greater detail in the discussion section of the companion paper and relevant references for further details are included.**

The distinction made between various simulation approaches (Lines 120-140) should also clarify Flow-py is modelling the dense core of a mass movement not a powder cloud, as some (e.g. RAMMS-Extended) do, which would obviously affect the total runout delineation.

**In lines 120-140 we have not yet introduced Flow-Py or RAMMS. We are only comparing process-based models with data driven ones. We do not think it makes sense to include the information that Flow-Py is a dense core model in this paragraph. We have now incorporated it in lines 152-157, where we introduce Flow-Py and compare it with the previous TauDEM model used. We do not think it adds any value to make this distinction here and comparing it to RAMMS-Extended.**

**We will include a sentence in the limitations section stating that the potential damage caused by powder blasts and the associated further runout distances are not captured by Flow-Py, and hence, are not considered in AutoATES v2.0.**

The justification for removing roughness from the PRA calculation is not compelling enough (4.1.3). Including parameters that assess terrain shape over several spatial scales (3x3 window vs 10x10 etc.) can help distinguish between steep features surrounded by other steep terrain and steep terrain that is surrounded by flat terrain (e.g. in a runout zone).

This  is a good idea, and it was a part of AutoATES v1.0. First, how roughness is applied at the PRA from Veitinger et al. 2016, its combined with a snow depth input. As the snow depth increases, the roughness factor loses its weight. In the PRA manual from Veitinger, he explains that the roughness is in effect being neglected if the DEM is coarse. For a 30 m DEM, we would need >15 m of snow to "smooth" out the roughness. This has two negative impacts for us; (1) we must use some sort of snow depth input, which would make the model reliant on snowpack factors, something ATES should not be. We are only interested in the terrain, and (2) we are not measuring roughness anymore, we are measuring the roughness of larger terrain features such as e.g. undulating terrain. A high roughness would make the PRA less likely to avalanche, while the "roughness" of larger terrain features, or undulating terrain would make it more complex to navigate in. So, by applying the roughness at a different scale than originally intended, the weight should be inverted being more complex terrain, not less likely to be a PRA. This is actually a good idea for the AutoATES algorithm as an improvement in identifying undulating terrain as being more complex, but we do not think it belongs within the PRA component in our use case.

One example of this is something we found in the PRA´s from AutoATES v1.0 in Norway. Areas with very undulating terrain was identified as "rough" terrain and got the value 0 in the Cauchy function because it was too "rough" for an avalanche to release. However, because the roughness was at mountain feature scale, and not the roughness of the ground, the whole area including several relevant PRA´s got the value 0. The Fuzzy operator is very vulnerable for 0 values, if one of the Cauchy membership values is 0, avalanches cannot release. We could probably get around this by modifying the Cauchy membership values, but as argued above, we believe it makes sense to keep the roughness at our intended scale (10-30 m DEM´s).

We will expand on this in section 4.1.3.

The Dev6 Post-forest-classification is not clear enough. Please explain Lines 550-551 where sections had to be reclassified. How should other users approach this manual intervention?

The manual intervention for this is to include the post-forest-classification step. So, users would not have to do any manual intervention other than using our post-forest-classification step.

What we mean by this is that including forest density in the PRA was not enough, there were still many areas with dense forest in runouts being classified higher compared to the benchmark consensus map. To solve this issue, we added the post-forest-classification.

With regards to ATES comparison (Section 3), could you provide an uncertainty estimate from the area mis-classified as percent of total area mapped?

We are not 100% sure whether we understand what the reviewer is asking for here. This metric could be read from the confusion matrices showing the percent of pixels correctly classified for different classes (as seen in Table 4).

"Avalanche zoning" (used several times) is usually applied to engineering situations. "Avalanche terrain classification" is clearer.

The same comment was made by Reviewer #1, we will use "avalanche terrain classification" throughout the updated manuscript as suggested.

**Technical corrections**

Lines 17-18: This note seems out of place here. Check with style guide, but I would suggest introducing the interchangeable use of 'model' and 'algorithm' in the introduction. Ideally you would only use one term in the paper if they are supposed to mean the same thing.

**We will remove this sentence.**

Line 25: These sites will be unknown to many – generalize e.g., '…two study sites in W Canada'

**We will change the sentence as suggested.**

Line 27: These values are very precise for an abstract.

**We received the same comment from Reviewer #1, and we will round these values to whole integers.**

Line 28: "Large-scale mapping" refers to a small geographic area. I think you mean a regional-scale mapping approach.

**We will change to "regional-scale mapping".**

**Introduction**

Lines 42-43: Sentence "This results in…" needs to be re-phrased/placed in more context.

**This sentence will be rephrased: "The spatial and temporal variability in the snowpack increases the complexity and results in a wicked learning environment, where feedback is not always reliable (Fisher et al., 2022)."**

Lines 110-111: Explain why Veitinger PRA output is 0-1 (based on fuzzy membership approach). And explain the terrain parameters used in these two approaches.

**We will explain this further, and add which parameters are used for both approaches.**

Lines 120-121: Cite some examples.

**We will add citations to Christen et al., 2010; Sampl and Zwinger, 2004; Tarboton, 1997; D'Amboise et al., 2022 as suggested.**

Line 152: briefly explain what "intensity of avalanches" means here, as this is not self-evident nor the same as impact pressure, for example, that comes from the depth-averaged modelling approach.

**Our understanding is that the $Z_{delta}$ parameter in Flow-Py is similar to potential velocity. This gives a sense of the total potential magnitude of an avalanche for a given path. While it is not the same an impact pressure, it is the same idea to try and estimate the destructive potential based on velocity.**

**We will add a sentence on this to the revised manuscript.**

**Section 2**

Lines 198-200: Provide citation for this statement.

**We will add a citation to Bebi et al. 2009.**

Line 209-210: Provide citation for this statement.

**We will add a citation to Bebi et al. 2009.**

Line 213: Stem density is stems per ha?

**We will rephrase this sentence so it states stems per hectare, this was a writing error.**

Lines 245-246: More justification, clarification is needed on scale issues with roughness.

**We will add a sentence referencing to the original PRA algorithm being made for a 2 m cell size. We´ll also add some discussion around why we did this in section 4.1.3 as per comment above in general comments.**

Lines 247-248: Please explain these new membership functions.

**We will rephrase this sentence to explain these new membership functions.**

Lines 248-29: Please explain what you mean by the values could be fine-tuned. Did you choose not to fine-tune them, or do you mean they could be fine-tuned based on local conditions in other regions? This is explained better in the caption to Figure 2 than in the text.

**We will explain why these values could be fine-tuned for local conditions / specific datasets.**

Figure 2: Please explain the a,b,c legend in the caption.

**We will explain a, b, and c in the caption for Figure 2.**

Line 292: Previous model here referring to AutoATES v1?

**We will rewrite this so it says AutoATES v1.0**

Table 1: are the Range values for Overhead exposure in number of cells or area? Please specify and use area rather than cells on account of varying spatial resolution of DEMs.

**Thank you for making this comment, we discovered that the values provided was from a previous iteration of AutoATES. The values should be 5 and 40 instead of 50 and 350. This input parameter is based on the degree of overhead exposure, relative to the maximum possible exposure based on the max $z_{delta}$ input parameter for Flow-Py. We have decided to scale this value to the range of 0-100 to simplify the interpretation, with 100 representing the maximum possible overhead exposure due to avalanche runout. We will specify this in section 2.4.2 and update Table 1 with the correct values.**

Table 2: Provide units for basal areas.

**The units are meters squared per ha, however there is a little more detail to be considered (see below). We will update Table 2 to include this information. The long answer is described/referenced in the companion paper.**

**Basal area is the total cross-sectional area, at breast height, of all living trees visible to the photo interpreter in the dominant, codominant, and high intermediate crown positions for each tree layer in the polygon. The polygon is visually conceived as a whole. This impression is converted to basal area (square meters per hectare) by estimating stand structure, species composition, form factors, height by species, stems per hectare, site, and uniformity.**

Table 2 Caption: Please clarify what you mean by encoding is identical for all three forest density types when we the encoding values are different as presented in the table under Encoding column.

**We will clarify how this encoding works in the revised manuscript.**

Table 3: I don't see the red coloured text is needed here. If you really like it, make note of it in the caption.

**We will remove the red-colored text in the revised manuscript.**

Lines 337-338: This sentence needs a citation at minimum. How is accuracy determined here—based on the reference expert map?

**We will remove the sentence as it does not fit within the methods section.**

**Results and validation**

Lines 363-364: More detail is needed on the study sites used as refence/benchmark. If this is covered in another paper, please refer reader for more detail.

**We provide this information a few sentences later: Sykes et al. (2024) provide an in-depth description of how the benchmark maps were developed.**

Figure 3: state this figure was adapted from Sykes et al., 2024. The inset maps are too small to be useful. Ensure updated manuscript has higher-resolution image. What are data sources (e.g. forest, DEM) and image credits?

**We will state that this figure is adapted from Sykes et al. in the revised manuscript. The figure will be remade with a larger inset map.**

**The forest data is British Columbia Vegetation Resource Inventory basal area, with the ALOS 30m DSM as the hillshade map. Images were provided by research collaborators at Parks Canada.**

Figure 4: Ensure high-res image is included in the revised manuscript.

**We will provide a high-res image for the revised manuscript.**

Lines 384-385: Please specify how you prepared the ATES benchmark. Was this already in a 10m raster format? Was your test conducted with a 10m DEM? Was it the same DEM used / available for the ATES benchmark? This may be better placed in Section 3.

**The ATES benchmark maps were created as polygon data sets manually delineated by human experts. Three individual maps were initially generated, then the three mappers collaborated to produce the final map. The vector data set was then rasterized using the resolution of the input DEM data set for comparison with the AutoATES output. We will expand on this around lines 384-385.**

Table 4: Adjust column headers to fit with accuracy scores.

**We will correct the column header in the revised manuscript.**

Figure 5 and Lines 398-404: I am not sure how useful this section is given that AutoATES v1 does not have the Extreme category. Unless it was compared to a previous version of ATES with only three categories?  Please clarify.

**We do not have any strong feelings for this visual representation of the two models, and the Reviewer has a point that they are not directly comparable as they have a different number of ATES classes. We will remove this figure with associated text from the revised manuscript.**

**Discussion**

Lines 466-469. Consensus matrices have not been introduced yet and this sentence needs re-phrasing.

**In this sentence we meant confusion matrices which have been introduced before. We´ll fix this in the revised manuscript.**

Section 4.1.3 heading does not match discussion of roughness and should be separated from forest density.

**We will make a new section for roughness and expand this discussion as per general comment above.**

Line 565: VRI introduced for the first time here. It should be introduced earlier.

**We will introduce this in the model development section in the revised manuscript.**

Line 575: rephrase 'reevaluate'

**This sentence will be rephrased.**

---

## Referee Report (RR1)

Round 2 review for NHESS- 2023-114

**General feedback**

The authors have done a good job largely addressing reviewer comments and improving the manuscript. There are still some organizational issues with the writing that need to be addressed, in particular in the introduction. Find examples below. There are some small writing style changes that still need attention, which I have also highlighted below, and one general point still needs refinement. Once these have been addressed I think the paper is ready for publication.

It was stated in the response to reviewer comments that you will clarify that Flow-Py is considering the dense core flow of the avalanche and any potential runout associated with a powder cloud is not considered. The manuscript has not (as far as I can tell) been updated to reflect this. I don't think it needs elaboration or a discussion of the powder cloud, but rather a simple statement in the introduction of Flow-Py to clarify it models the dense core, as well as how the AutoATES v2.0 is based on dense core runout extents.

**Specific points to be addressed**

Lines 23-26: Sentence should be split to make interpretation easier.

Line 30. 'Large-scale mapping' is used, though you said you would use 'regional-scale mapping' in the response to reviewer. It is fine to use 'large-scale' to discuss a something covering a large area, but 'large-scale mapping' specifically refers to map scale (and covers small areas). Maybe referring to large-scale ATES classification and avoiding use of the 'mapping' term solves the problem.

Section 1 has several small paragraphs (including lines 79-81 which is a single sentence) that needs combining.

Section 1.1 has duplicate sentences/information (lines 95- and lines 103-). The same information is again repeated at the start of Section 2.4.1. where your customized implementation of PRA is introduced.

Section 1.1 needs reorganizing to aid interpretation. For example the PRA is discussed in specific terms before being introduced in a general sense. The sentence beginning in line 113 should come at the start of the section before discussing the various methods of implementing a PRA with terrain and forestry data.

Line 123: Citation does not need to be italicized.

Line 130: Typo 3-dimensional

Lines 131-133: While the computational power required to apply the process-based models over large areas is a factor, it is being done at regional scales (e.g., Bühler et al. 2022).

Table 4: Square meters looks to be using subscript instead of superscript $m^2$

Table 4: Inconsistent use of comma as thousands separator.

Line 402: Typo ', model, or model'

Lines 438-439: Typo, end of sentence unclear.

Line 535: Avoid conjunctions; "We don't know why this is…" could be "The reason for this is unclear…"

Line 547: Typo 'boundaries'

Line 597: This is the first mention that Flow-Py is computationally 'heavy'. One argument for using Flow-Py presented in section 1.2 is that it is more computationally inexpensive compared with process-based models. I suggest re-phrasing the sentence or adding specifics for this limitation to have more utility to potential users. For example, reporting the run-time and computer specs for the benchmark sites.

Line 560: Avoid emotive word choice ("Blindly applying the parameters…") by deleting 'Blindly'

Line 604: Same comment as above. Sentence could read "Users should not adopt the input parameters provided in the paper without thorough testing."

Line 617: Typo, missing 'of'

**References**

Bühler, Y., Bebi, P., Christen, M., Margreth, S., Stoffel, L., Stoffel, A., Marty, C., Schmucki, G., Caviezel, A., Kühne, R., Wohlwend, S., and Bartelt, P.: Automated avalanche hazard indication mapping on a statewide scale, Nat. Hazards Earth Syst. Sci., 22, 1825–1843, https://doi.org/10.5194/nhess-22-1825-2022, 2022.

---

## Referee Report (RR2)

[revised manuscript text omitted]
. (2016) due to its continuous raster output ranging from 0 to 1. The model uses windshelter, roughness, slope angle and forest density as inputs. However, the forest density is only processed as a binary input, meaning that the input is either forested or non-forested. If an area is defined as forested, it is not processed by the PRA model and defined as a non-PRA. Sharp (2018) improved the PRA model by incorporating forest density as a parameter

in the fuzzy logic operator, making the interaction of forest density dynamic and equally important compared to roughness, slope angle and windshelter.

In AutoATES v1.0, Larsen et al. (2020) utilized the PRA model by Veitinger et al. (2016), which outputs a continuous range of values between 0 and 1. This model considers factors such as windshelter, terrain roughness, slope angle, and forest density. Originally, forest density was only a binary input, effectively categorizing areas as either 'forested' or 'non-forested'. In the binary approach, any 'forested' area was not further processed by the PRA model and was simply labeled as non-PRA. In 2018, Sharp improved the PRA model by including the forest density parameter in what's known as a fuzzy logic operator. Fuzzy logic, unlike binary, does not restrict inputs to yes-or-no values; instead, it allows for degrees of truth. For instance, instead of an area being classified as simply 'forested' or 'not forested,' it could be 'somewhat,' 'mostly,' or 'completely' forested. This method acknowledges the nuances 
[revised manuscript text omitted]

785

---

## Author Response (AR2)

Round 2 review for NHESS- 2023-114

**General feedback**

The authors have done a good job largely addressing reviewer comments and improving the manuscript. There are still some organizational issues with the writing that need to be addressed, in particular in the introduction. Find examples below. There are some small writing style changes that still need attention, which I have also highlighted below, and one general point still needs refinement. Once these have been addressed, I think the paper is ready for publication.

Thank you for your thorough review. We found your comments very constructive. We have tried our best to revise our manuscript in accordance with your suggestions. We hope that you will find that the responses sufficient for publication. If you should find that our responses are insufficient, we welcome further feedback on how to improve it.

It was stated in the response to reviewer comments that you will clarify that Flow-Py is considering the dense core flow of the avalanche and any potential runout associated with a powder cloud is not considered. The manuscript has not (as far as I can tell) been updated to reflect this. I don't think it needs elaboration or a discussion of the powder cloud, but rather a simple statement in the introduction of Flow-Py to clarify it models the dense core, as well as how the AutoATES v2.0 is based on dense core runout extents.

We have included a statement that Flow-Py is a dense core model (lines134-135) and that AutoATES v2.0 is based on dense core runout extents (lines 266-267).

*"Recently, D'Amboise et al. (2022) presented a new customizable simulation package (Flow-Py) to estimate the runout distance and intensity of dense core avalanches (not considering powder clouds)."*

*"The Flow-Py model developed by D'Amboise et al. (2022) is used for the avalanche simulation of the potential track and deposition area. Flow-Py is a dense core model, thus AutoATES v2.0 is based on dense core runout extents and does not consider powder clouds."*

**Specific points to be addressed.**

Lines 23-26: Sentence should be split to make interpretation easier.

We have split this sentence in the revised manuscript.

Line 30. 'Large-scale mapping' is used, though you said you would use 'regional-scale mapping' in the response to reviewer. It is fine to use 'large-scale' to discuss a something covering a large area, but 'large-scale mapping' specifically refers to map scale (and covers small areas). Maybe referring to large-scale ATES classification and avoiding use of the 'mapping' term solves the problem.

We will refer to large-scale ATES classification as suggested above.

Section 1 has several small paragraphs (including lines 79-81 which is a single sentence) that needs combining.

We have combined lines 79-81 with the paragraph above. We have also combined lines 59-69.

Section 1.1 has duplicate sentences/information (lines 95- and lines 103-). The same information is again repeated at the start of Section 2.4.1. where your customized implementation of PRA is introduced.

Thank you for discovering the duplicate information. We have removed lines 93-103 as we had rewritten the paragraph below (lines 103-) without removing the old text by mistake. We have also removed the repeated information from the start of Section 2.4.1.

Section 1.1 needs reorganizing to aid interpretation. For example the PRA is discussed in specific terms before being introduced in a general sense. The sentence beginning in line 113 should come at the start of the section before discussing the various methods of implementing a PRA with terrain and forestry data.

We have moved the sentence from line 113 to the start of section 1.1. We also believe that removing the first erroneous paragraph makes this section more easy to interpret.

Line 123: Citation does not need to be italicized.

We have the italicized text to regular.

Line 130: Typo 3-dimensional
Typo corrected.

Lines 131-133: While the computational power required to apply the process-based models over large areas is a factor, it is being done at regional scales (e.g., Bühler et al. 2022).

We have added this sentence: Even though the computational power required to apply the process-based models over large areas is a factor, it could be done at regional scales (e.g., Bühler et al. 2022).

Table 4: Square meters looks to be using subscript instead of superscript m2

Typos corrected.

Table 4: Inconsistent use of comma as thousands separator.

Typos corrected.

Line 402: Typo ', model, or model'

Typo corrected.

Lines 438-439: Typo, end of sentence unclear.

We have rephrased the sentence: Initial attempts by Larsen et al., (2020) compared AutoATES v1.0 to available linear and spatial ATES ratings in Norway, however the validity of these ratings was uncertain because they were developed with limited peer-review and could be biased.

Line 535: Avoid conjunctions; "We don't know why this is…" could be "The reason for this is unclear…"

Thank you, we have changed the text to: The reason for this is unclear…

Line 547: Typo 'boundaries'

Typo corrected.

Line 597: This is the first mention that Flow-Py is computationally 'heavy'. One argument for using Flow-Py presented in section 1.2 is that it is more computationally inexpensive compared with

process-based models. I suggest re-phrasing the sentence or adding specifics for this limitation to have more utility to potential users. For example, reporting the run-time and computer specs for the benchmark sites.

We have rephrased this section as follows: In the context of large-scale ATES classification (e.g. Norway, 385,207 km2), Flow-Py becomes computationally heavy, which may present challenges when processing large datasets or applying the model in real-time applications. We executed the Flow-Py algorithm across all of Norway on an Amazon Web Services Elastic Cloud Compute Instance (AWS EC2 c6g.metal), which took 30 days to complete at a cost of $1,600. This could potentially limit the scalability and accessibility of the model for certain use cases and users with limited computational resources.

Line 560: Avoid emotive word choice ("Blindly applying the parameters…") by deleting 'Blindly'

We have deleted the word "blindly".

Line 604: Same comment as above. Sentence could read "Users should not adopt the input parameters provided in the paper without thorough testing."

We have deleted the word "blindly".

Line 617: Typo, missing 'of'

Typo corrected.

**References**
Bühler, Y., Bebi, P., Christen, M., Margreth, S., Stoffel, L., Stoffel, A., Marty, C., Schmucki, G., Caviezel, A., Kühne, R., Wohlwend, S., and Bartelt, P.: Automated avalanche hazard indication mapping on a statewide scale, Nat. Hazards Earth Syst. Sci., 22, 1825–1843, https://doi.org/10.5194/nhess-22-1825-2022, 2022